# Escalation vs. Early Intense Therapy in Multiple Sclerosis

**DOI:** 10.3390/jpm12010119

**Published:** 2022-01-17

**Authors:** Bonaventura Casanova, Carlos Quintanilla-Bordás, Francisco Gascón

**Affiliations:** 1Unitat de Neuroimmunologia, Hospital Universitari i Politècnic La Fe. València, la Universitat de València, 46026 Valencia, Spain; carlosqb@gmail.com; 2Unitat de Neuroimmunologia, Hospital Clínic Universitari de València, 46010 Valencia, Spain; kokogascon@hotmail.com

**Keywords:** multiple sclerosis treatment, escalating strategy, he-DMT, diseases modifying therapies, early intense therapy

## Abstract

The treatment strategy of multiple sclerosis (MS) is a highly controversial debate. Currently, there are up to 19 drugs approved. However, there is no clear evidence to guide fundamental decisions such as what treatment should be chosen in first place, when treatment failure or suboptimal response should be considered, or what treatment should be considered in these cases. The “escalation strategy” consists of starting treatment with drugs of low side-effect profile and low efficacy, and “escalating” to drugs of higher efficacy—with more potential side-effects—if necessary. This strategy has prevailed over the years. However, the evidence supporting this strategy is based on short-term studies, in hope that the benefits will stand in the long term. These studies usually do not consider the heterogeneity of the disease and the limited effect that relapses have on the long-term. On the other hand, “early intense therapy” strategy refers to starting treatment with drugs of higher efficacy from the beginning, despite having a less favorable side-effect profile. This approach takes advantage of the so-called “window of opportunity” in hope to maximize the clinical benefits in the long-term. At present, the debate remains open. In this review, we will critically review both strategies. We provide a summary of the current evidence for each strategy without aiming to reach a definite conclusion.

## 1. Introduction

### 1.1. Multiple Sclerosis: A General Overview

Multiple sclerosis (MS) is a disorder of remarkable heterogeneity that affects the central nervous system. It is characterized by inflammatory attacks to the myelin and axons, and by neurodegenerative cascade that give rise to progression of the disease independent of the initial inflammatory activity [1,2].

Based on these immunopathogenic mechanisms we can find two clinical forms of onset: bout-onset and progressive-onset disease. Bout-onset multiple sclerosis (BOMS) is characterized by relapses and remission (hence, relapsing-remitting MS (RRMS)). More than 50% of patients with BOMS will develop after variable time sustained progression of disability independent of the relapses, and hence, will convert to secondary progressive MS (SPMS). On the other hand, progressive-onset MS (POMS), is characterized by a sustained worsening of the disability since the beginning of the disease (hence, primary progressive MS [PPMS]) [3,4].

Relapses and progression are the main determinants of disability in MS [5]. However, natural history studies show that once progression becomes clinically evident, disability is no longer determined by the presence of previous relapses, and therefore is independent of the clinical form of onset (either BOMS or POMS). In fact, relapses have been shown to be minor contributors of progression of the disease. Specifically, short time between the first and second relapse and a sudden increase in the relapse rate in the preceding two years have been linked to the risk of developing SPMS, and yet they have little or no impact on disability once the progression has started [6,7].

At this point, two concepts must be introduced: relapse-associated worsening (RAW); the determinant of disability in RRMS, and steady progression independent of relapse activity (PIRA); the main determinant of disability in SPMS and PPMS [8,9,10]. Two other important considerations must be considered: the uncertainty of the real disease onset, and the lifelong duration of the disease. The diagnosis of MS is made based on the presence of relapses or sustained progression of disability in conjunction with typical magnetic resonance imaging and/or cerebrospinal fluid features. Therefore, it is important to recognize that the diagnosis may not truly mark the real onset of the disease, but rather the time at which the disease becomes clinically apparent and fulfills our established criteria to minimize the risk of misdiagnosis. Furthermore, BOMS is diagnosed around thirty years old, and SPMS and PPMS around forty-five years old. Considering life expectancy may be reduced by 5–7 years on average, it means that the average disease duration will be of 50 years for BOPMS and 35 years for POMS. These facts must be considered when analyzing clinical trials, as the apparent clinical disease duration may not be representative of the real disease duration, (thereby acting as a confounding factor)*,* and because the follow-up times of these trials are too short to evaluate the impact over the long-term of the disease [11].

Continuing with the uncertainty surrounding MS, one of the main problems we face in treating MS is whether there is activation of the innate immune system, which has been linked to the progression of disability, from the onset of the disease or whether it occurs at some point as a result of dysregulation of the acquired immune system joint to the phenomenon of antigenic spread. There is currently no answer to this question, but very early initiation with induction drugs that reset the immune system, such as Alemtuzumab, have given the best results in terms of long-term progressive secondary progressive progression, which may indicate that intense early treatment could prevents activation of the innate immune system and triggering of the innate immune-dependent mechanisms responsible for the progression of the disability.

### 1.2. The Selection of the Objective in Clinical Trials

According to natural history studies and previous knowledge of the disease, the two main clinical endpoints chosen in phase III trials have been relapses and progression of disability [12]. When correlation between relapses and gadolinium-enhancing lesions (GEL) in the MRI was proven, it was also introduced as a surrogate marker of disease activity in phase II trials [13,14,15,16]. In fact, phase II trials have now replaced annualized relapses rate (ARR) with number of GEL as the primary endpoint, becoming the former a secondary endpoint, which has allowed trials to be shortened to six months. Meanwhile, phase III trials have maintained clinically defined primary endpoints: ARR for RRMS trials, and time to sustained progression confirmed in 3 or 6 months for PPMS trials [17].

Clinical secondary objectives in RRMS have been time to sustained progression of disability confirmed at 3 or 6 months. With the introduction of the MRI, T2 lesion number and volume, GEL number, and more recently, brain atrophy measures have been implemented in clinical trials [18]. However, it is important to consider the limited effect that these primary endpoints have in the natural history of the disease (i.e., development progressive course) and the potential confounding between relapse-related worsening and progressive disease [11]. This may explain the discrepancy observed between short-term efficacy of DMT on EDSS (mostly dependent on disability accumulation due to relapses) and the absence of effect to delay conversion to SPMS and to slow down progression of disability in PPMS and SPMS. An approach to deal with this problem has been to introduce the aforementioned concept of PIRA in clinical trials, as a way to distinguish progression of disability independent of relapses [19].

Also, scales other than EDSS such as the Multiple Sclerosis Functional Composite (MSFC), and other clinical and radiological variables, have been introduced to increase the sensitivity of progression of disability. However, uncertainty regarding the real clinical impact of these measures has limited their use [20,21].

Finally, care must be taken when assessing the clinical endpoint of a clinical trial. Firstly, the ARR may not harvest all the “focal” inflammatory activity. Secondly, ordinary brain MRI monitoring may fail to detect cortical lesions and ectopic meningeal follicles. Thirdly, progression measured by the EDSS is insensitive to minor clinical changes, especially when related to cognition [2].

For these reasons, treatment strategy in MS must be based on a judicious interpretation of the evidence from clinical trials. Special attention should be given to the actual results observed, as they may not reflect the reality of the disease and long-term effectiveness.

### 1.3. Classification of Disease Modifying Therapies

Before 1993, MS treatment was based on several immunosuppressive drugs, but it was with the approval of interferon (IFN) beta-1b (Betaferon^®^) that a new era of MS treatment began. Since then, nineteen drugs have been approved and four are already in the process of approval by the regulatory agencies. Obviously, each drug has a particular mechanism of action (MoA), level of effectiveness, and safety profile. According to the decrease in the ARR, DMT have been classified as moderate-intermediate efficacy and high-efficacy therapies (HET). IFN, glatiramer acetate (GA), azathioprine, and the newer orals drugs teriflunomide and dimethyl fumarte (dimethyl fumarate) are usually considered as being of moderate efficacy. Fingolimod, other sphingosine-1-phosphate (S1P) receptor modulators and cladribine are usually considered as intermediate efficacy drugs. Finally, monoclonal antibodies (MoAb) such natalizumab, alemtuzumab and ocrelizumab, together with mitoxantrone (an antineoplastic agent) are usually considered HET. Daclizumab, a MoAb, is no longer considered as a treatment for MS due to severe and unacceptable secondary effects [22,23,24,25,26,27,28,29,30,31,32,33,34].

Aside to this classification, DMT have been also classified as “first-line” therapies and “second-line” therapies. The former, DMTs of moderate efficacy but low side effects profile such IFN, GA, teriflunomide and dimethyl fumarate are usually included. In the USA, fingolimod and cladribine are also considered as “first-line” DMT. “Second-line” therapies include the MoAb and mitoxantrone. The use of these terms is applicable for the treatment of RRMS, but not for PPMS and SPMS [35].

Classification of DMT into “lines” of treatment has been the most popular one, and this has determined the escalation-based treatment approach, despite the absence of clinical evidence and disparities in this classification. For example, fingolimod is considered as a “second-line” drug in Europe, but a first-line drug in the USA. On the other hand, dimethyl fumarate, despite having several cases of progressive multifocal leukoencephalopathy (PML), is considered a first-line therapy. Additionally, cladribine, which is approved in Europe for highly active RRMS, and in the USA also for relapsing forms SPMS, has not been robustly studied in either of these settings.

For more than 25 years, this absurd classification into lines of treatment that implies an escalation approach strategy has prevailed. The question arises as to whether this is the best approach when considering the long-term impact on the disease [36].

### 1.4. The Concept of Treatment Failure

The first double blind randomized placebo controlled clinical trial in RRMS, tested IFN-1b against placebo. After three years, the number of relapse-free patients was 17/123 (13.8%) in the placebo arm; and 27/124 (21.7%) in the treated-arm [37]. It was clear that almost 2/3 of patients had relapses despite treatment. Consequently, three lines of investigation were established: to identify non-responders; to define baseline characteristics of non-responders; and to study the consequences of a suboptimal response. All these lines of investigations prompted a definition of treatment failure (TF) or suboptimal response (RSO). Waubant et al. were the first authors to define TF, based on the relapse rate, and defining TF as an ARR similar to the previous year [38]. Rio et al. used different criteria to define TF, as it included progression of EDSS, and a combination of clinical activity and MRI outcomes. However, progression of EDSS was a confounding factor, as IFN was not intended to treat progression. Sormani then modified these criteria (hence, the modified Rio-Sormani score), which remain as the most widely accepted definition of TF in current studies [39,40].

It is important to highlight that that the modified Rio-Sormani score has only been validated with IFN-1b treatment. If we want to apply these criteria in current clinical practice, we must consider the real value of a drug and the consequences of TF. For example, if a given drug is not intended to treat progression, it is reasonable that TF should not be considered when treatment has no effect on progression.

In agreement with this idea, it does not seem reasonable to consider TF when a progressive increases in disability are demonstrated under a determinate treatment, if the treatment have no effect over progression, at the same time, the presence of some inflammatory activity in form of relapses and/or MRI activity, are expected.

The short duration of clinical trials and observational studies has set the focus on the short-term effect of DMTs over disability, leading to the escalation strategy. However, long-term studies show DMT have a scarce effect on the risk of conversion to the SPMS and on disability once the progressive phase has started. This raises the question as to whether an escalation approach therapy is really appropriate [9,41].

Furthermore, if we follow the principle primum non nocere (“first, do no harm”), we should be cautious when escalating therapy, as HET initiation years after the disease increase the likelihood adverse events, but still do not change the long-term prognosis of the disease. HET have shown to have a greatest effect on the risk of conversion to SPMS when initiated early in the disease, during the so-called “window of opportunity”. Still, early treatment with HET may expose young, healthy individuals with minor disability to serious side effects. Thus, the real debate should be whether the risks of early initiation HET outweigh the risks over the long term, and not where early intense therapy is “more efficacious” than escalation therapy.

## 2. Escalation Therapy

### 2.1. Definition

Escalation therapy must be clearly defined to compare studies and management strategies consistently. A European survey about MS management showed that treatment escalation or initiation based on relapses or new T2 lesions varied significantly between different countries, territories and even at institutions themselves [42]. There was a high agreement in switching to a HET when a patient experienced either two relapses, 5–8 new T2 lesions or two gadolinium enhancing lesions within a year [43]. However, this threshold is probably too high since these patients have a high probability to develop SPMS in the next two years. Hence, to evaluate the impact of treatment escalation, studies must clearly define previous DMTs, time evolution of the disease, and most importantly, the reason for treatment escalation. Otherwise, these studies may lead to paradoxical results, as was the case in the based on one of the largest registries of patients with multiple sclerosis, the MSBase registry, which used propensity score methodology, showed that patients starting on HET (fingolimod, alemtuzumab or natalizumab) had lower probability of conversion to SPMS when compared to patients starting on GA or IFN. In this series, time to treatment initiation was 6.5 years for HET vs. 5.1 years for GA or IFN. Furthermore, the authors reported a lower risk of conversion to SPMS when GA or IFN was started within 5 years versus later. In fact, treatment escalation after 5 years of evolution did not have a clear effect on the probability of converting to SPMS [43].

These evidence highlights the importance in defining to whom and to what DMT is being changed, as this is they are the only way to obtain clear conclusions from the escalation-based treatment strategy. It is not an academic question, as data shows that timing of treatment is crucial to impact significantly the probability to convert to SPMS, which might be related to effects of an aging immune system (i.e., immunosenescence) and loss of a potential “window of opportunity”, which evidence suggests is limited to around 5 years.

### 2.2. Trials That Support Escalation Therapy

The currently approved HET are: natalizumab, fingolimod, alemtuzumab, ocrelizumab and cladribine. However, only fingolimod, alemtuzumab and ocrelizumab have been studied in appropriate clinical trials and have class A evidence of superiority with respect to other first-line therapies.

#### 2.2.1. Fingolimod

The TRANSFORMS trial, fingolimod was assessed against IFN beta 1-a for one year in patients with RRMS between 18 and 55 years [44]. Patients had to have a relapse within the previous year, or two relapses within the previous two years, and a EDSS score between 0 and 5.5. 1292 patients were randomized: 426 to fingolimod 1.25 mg once a day; 431 to fingolimod 0.5 mg once a day-; and 435 to follow with IFN beta 1-a 30 μg i.m. weekly. The main clinical results were a reduction of the ARR of 0.16 in the 0.5 mg fingolimod arm, vs. 0.33 in the IFN beta 1-a arm; 82.6% of relapse-free patients in the 0.5 mg fingolimod arm vs. 69.3% in the IFN beta-1-a arm; and no differences with respect to progression of disability after one year. The main criticism of this trial (and other subsequent trials) is that the “active” treatment was continued even during a suboptimal response. Moreover, 45% of patients already had been treated with a previous DMT. Therefore, is difficult to draw conclusions about the effect of a treatment when is compared to another that has previously failed in these patients. Even so, the main clinical endpoint (progression of disability) was not met. Moreover, the extension trial at two years, despite having good results in MRI variables, still did not show any significant differences in progression of disability.

#### 2.2.2. Alemtuzumab

The second HET that was explored against an active comparator in naïve patients or that failed to a previous treatment was alemtuzumab. This was assessed in the two-phase III clinical trials: the CARE-MS I and the CARE-MS II. In the former, 581 naïve patients with RRMS were randomized in a 2:1 proportion to receive either alemtuzumab 12 mg or Rebif-44 3 days a week and followed up for two years. The main results were a reduction of the ARR of 0.18 in the alemtuzumab arm vs. 0.39 in the Rebif-44 arm, and 77.6% of relapses-free patients in the alemtuzumab arm vs. 58.7% in the Rebif-44 arm. There were no differences in sustained accumulation of disability confirmed over 6 months between groups, possibly due to the low baseline disability of patients. In the CARE-MS II trial, 636 RRMS patients with at least one relapse while on GA or IFN beta were randomized in 2:1 proportion to receive alemtuzumab 12 mg or Rebif-44. After two years, 65.4% in the alemtuzumab 12 mg arm remained relapse-free vs. 46.7% in the Tebif-44 arm. Unlike CARE-MS I, this trial showed a lower rate of sustained confirmed progression at 6 months in the alemtuzumab arm (13% vs. 20%). There was also a positive effect over the MSFC score and MRI measures. Hence, CARE-MS II, which used a population with higher inflammatory activity (previous treatment failure) and higher EDSS, did show a clear effect on disability accumulation [45,46].

#### 2.2.3. Ocrelizumab

The development phase III program of ocrelizumab in RRMS was done in two simultaneous and identical trials: OPERA-I and OPERA-II. Inclusion criteria required at least one relapse within the previous 2 years. These trials randomized a total of 1656 patients in a 1:1 ratio to receive ocrelizumab or Rebif-44. 71% and 73% were treatment naïve. The main results in the pooled analyses were a reduction in the ARR (0.16 in the ocrelizumab arm vs. 0.29 in the Rebif-44 arm), and a reduction in the proportion of patients reaching disability progression confirmed at 12 weeks (9.1% vs. 13.6%, respectively). However, subgroup analyses suggested that the reduction of progression of disability was not significant among the 224 patients that were previously treated with a DMT, despite having a positive effect on the ARR. Similarly, subgroup analyses showed that the reduction of progression was not significant when considering patients with a body mass index of 25 or more. This example of “tortured-data”, seems to suggest ocrelizumab may be more effective in lean than overweight patients [47,48].

### 2.3. Summary

The efficacy of escalation to HET has been evaluated in a myriad of observational studies, but level A evidence supporting this strategy is scarce. Although evidence shows that escalation therapy is useful to abrogate inflammatory activity, it has only showed a modest effect over the progression of disability. In fact, the longest observational studies still show that this strategy is futile to prevent conversion to SPMS. However, this does not prove that this strategy is not valid. We might consider it as a useful therapy to reduce relapses and the progression of disability in the beginning of the disease. Data suggest that beyond four or five years, the effect on relapses maintains, but not on accumulation of disability.

## 3. High-Efficacy Therapy

### 3.1. Definition

High-efficacy therapy (HET) refers to agents that have a greater impact on inflammation compared to moderately effective therapies [49]. Therefore, the classification of DMT as high-efficacy is based on favorable outcomes from clinical trials comparing that treatment usually to traditional DMTs (mostly inflammatory outcomes such as relapses and new lesions, although some experts prefer to evaluate lack of disability too) [49].

There is an agreement to consider natalizumab, antiCD20 therapy (rituximab and ocrelizumab), alemtuzumab, mitoxantrone, cyclophosphamide and autologous stem cell transplantation as HET. However, there is not a consensus regarding sphingosine-1-phosphate receptor modulators such as fingolimod (some consider it an intermediate efficacy therapy) and cladribine (as it has been compared to placebo, although it probably has an induction effect) [49] (Table 1).

As opposed to escalation, where treatment starts with a low-risk and lower-efficacy treatment and only moves on to a more aggressive treatment if the ongoing approach fails, early aggressive therapy or Early Intensive Therapy (EIT) considers starting high-efficacy treatment earlier in MS, mostly initially since its onset, to maximize the potential for preventing disability progression over time, assuming a higher-risk profile of adverse events [48,50].

Many consider that EIT as the best way to achieve long-term outcomes for people with MS, based on the following rationale: the ability to predict disease course at onset is limited, conventional imaging underestimates ongoing damage, irreversible nervous damage occurs very early and once neurological function is lost it cannot be regained. MS is rarely benign over the long term. Long term follow-up studies reveal the risks of undertreatment. Safety profile of some HET may not differ from low-efficacy treatment and it is mainly early intervention that might substantively alter disease course and prevent irreversible progression, whereas later treatment might not confer much benefit [3,4,5,6,7,8,9]. Therefore, EIT is based on using highly effective treatments starting early, while on the therapeutic window, where they are more effective than when started later on the disease course in the escalation approach [51,52,53,54,55,56,57] (Table 2).

However, when considering EIT, two main approaches arise. Firstly, Induction treatment (IT), also referred to as immune reconstitution therapy which is based on the use of HET with a sustained biological effect in naïve patients, followed or not by long-term maintenance treatment (generally with immunomodulatory agents) and secondly, sustained HET, which is based on the use of HET continuously, as their effect wanes when interrupting treatment [50,51,52,53,54,55,56,57]. Induction treatment includes mitoxantrone, cyclophosphamide, stem cell transplantation, alemtuzumab and cladribine, whereas the potential inductive effect of antiCD20 therapy is mild and of natalizumab and fingolimod is null (as their withdrawal is associated with reactivation of the disease) [50,51,52,53,54,55,56]. Induction treatments usually are associated with a higher risk profile but shorter in time as their administration is not sustained, while the use of continuous HET is associated with a risk profile sustained overtime. The rationale for induction therapy is to influence the inflammatory phase and to avoid the subsequent chronic phase resetting the immunological system to prevent the phenomenon of epitope spread [50,51,52]. The risk associated with treatment that is judged acceptable may vary with disease severity, however, disease severity might be underestimated, specially early at onset, and treatments are less effective as disease evolves [51,54,55,56]. It is known that MS patients treated early do better than those in whom treatment is delayed, but regarding the question does the potency of DMT truly matter, recent observational studies show better long-term outcomes on disability accumulation and risk of conversion to SPMS with EIT than escalation [57].

### 3.2. Results over Inflammatory Activity, Progression, and Safety

Natalizumab’s original trial, AFFIRM study, showed a 68% reduction in ARR at year 1, 42% relative risk reduction in disability progression at 2 years, 83% reduction of new T2 lesions and 92% reduction in contrast enhancing lesions compared to placebo. The REVEAL study compared natalizumab to fingolimod with a lower cumulative probability of relapse and gadolinium-enhancing lesions 70% lower in the natalizumab group. Several observational studies comparing naive patients treated with natalizumab vs. injectables DMT have shown greater reductions in ARR and disability accrual, and others, when comparing escalation to Natalizumab to those switching to fingolimod have shown higher rates of NEDA with natalizumab. The TOP study at 5 years reported lower ARR in naive natalizumab patients than those who escalated to natalizumab from prior DMTs. Natalizumab main risk are infusion related reactions and PML risk [49,63].

Regarding Alemtuzumab, phase II CAMMS223 in naive RRMS showed better results on relapses, disability accumulation, MRI activity and atrophy compared to interferon at 3 years. In the CARE-MS I study with naive RRMS alemtuzumab reduced significantly ARR and MRI activity at 2 years but not disability progression compared to interferon-beta-1A, and in the CARE-MS 2 study with RRMS who failed to previous DMT, alemtuzumab reduced the ARR, MRI and disability progression at 2 years compared to interferon. Extension studies up to 6 years showed sustained benefits of alemtuzumab on clinical and MRI activity and progression of disability in a great proportion of patients, and interestingly the conversion rate to SPMS at 6 years was very low (3%). A cohort study comparing alemtuzumab effectiveness to natalizumab, fingolimod and interferons, up to 5 years, revealed similar reductions on ARR for alemtuzumab and natalizumab, but superior to fingolimod and interferons, however natalizumab seemed better than alemtuzumab in enabling recovery from disability. Alemtuzumab safety risks include infusion reactions, stroke and arterial dissection, severe infections including opportunistic ones such as herpetic and Listeria monocytogenes, and secondary autoimmune disorders (thyroid disorders, idiopathic thrombocytopenic purpura, and nephropathies among others) [49,50,63,64,65,66].

Cladribine was studied in the 2-year placebo-controlled phase 3 study CLARITY, with lower relapse rates, lower risk of 3 months sustained disability progression and significant reductions in brain lesions counts. Moreover, the 2-year extension study of CLARITY showed that patients that received cladribine during the core study followed by placebo during the third and fourth year had sustained benefits in terms of activity and progression (similar to 4 years with cladribine treatment). However, Cladribine may be slightly less effective than other HET, as an observational study revealed a significant reduction on ARR with cladribine compared to medium efficacy therapies and a similar reduction compared to fingolimod, but a lower reduction compared to natalizumab. Cladribine safety issues are related mainly to lymphopenia and herpes zoster infections [49,50,64].

Rituximab is used off-label in MS. Rituximab compared to placebo in a phase II trial showed a lower risk of relapse and greater reductions on MRI activity. Several observational studies, especially from Sweden, have confirmed these results. Moreover, an observational study revealed that switching from natalizumab (due to JCV positivity) to rituximab was related to lower clinical and MRI activity compared to switching to fingolimod. In a comparative study with a 4-year follow-up, initial treatment with rituximab demonstrated a significant lower rate of relapses and MRI activity compared to injectable DMTs and dimethyl fumarate, with a tendency for lower relapse rates compared with natalizumab and fingolimod. The OPERA I and II phase III studies compared ocrelizumab to interferon-beta-1a in RRMS patients with greater reductions in ARR, MRI activity and progression of disability at 3 and 6 months with ocrelizumab. The most common side effects of anti-CD20 therapy are infusion reactions and infections (including cases of herpes zoster, hepatitis B reactivation and PML), although bone marrow suppression and neutropenia have been described [66,67].

Mitoxantrone was compared to placebo in a French-British randomized controlled trial and to Interferon-beta in a 3-year pivotal trial, and was related to a significant lower relapse rate, MRI activity and disability worsening. Another study compared induction with mitoxantrone followed by glatiramer acetate maintenance therapy vs. glatiramer acetate, with a significant reduction on ARR and MRI activity in the first group. Long-term mitoxantrone effectiveness has been studied up to 5–10 years of follow-up with significant results on reduction of disability worsening, compared to medium efficacy DMTs, especially when followed by platform treatment maintenance. The risk of severe adverse events such as heart failure or leukemia or amenorrhea make mitoxantrone a less suitable treatment option nowadays [49,50,62,64].

Regarding Cyclophosphamide, the two-year randomized clinical trial of cyclophosphamide followed by interferon vs. interferon alone showed a significant reduction in clinical and MRI activity, and an observational study using induction with cyclophosphamide followed by maintenance therapy with glatiramer acetate showed similar results. However, its safety profile mainly related to infections and hemorrhagic cystitis and bladder cancer have reduced its use nowadays [50,62].

Autologous hematopoietic stem cell transplantation (AHSCT) has been used in aggressive RRMS. The phase II ASTIMS trial demonstrated AHSCT was superior to mitoxantrone reducing relapse rates and MRI activity without differences in the progression of disability between groups. An observational study showed an important proportion of progression-free survival at 5 years of follow-up with AHSCT, with better outcomes with lower baseline EDSS. Another observational study revealed AHSCT is more suitable for aggressive RRMS as none of the RRMS experienced worsening of disability after a median follow-up of 5.4 years while 22.6% of SPMS experienced disability worsening. However, safety risks of AHSCT including infections and mortality, make AHSCT suitable only for aggressive RRMS patients refractory to high-efficacy conventional therapies and active disease with potential for disability accumulation. The BEAT-MS (Best Available Therapy Versus Autologous Hematopoietic Stem Cell Transplant for Multiple Sclerosis) study is a 6-year ongoing study currently investigating AHSCT versus high-efficacy DMTs (natalizumab, alemtuzumab, ocrelizumab, or rituximab) with a primary endpoint of relapse-free survival up to 36 months [49,50,68].

Interestingly, a recent Norwegian observational study has compared the short-term effect of initial HET (with natalizumab, fingolimod and alemtuzumab) vs. medium efficacy treatment. Initial HET was associated with a greater proportion of NEDA at years 1 and 2 compared to initial medium efficacy treatment (OR 3.9, *p* < 0.001, at year 1) [69].

### 3.3. The Importance of Long-Terms Outcomes. Analysis of the Comparative Studies: Escalation vs. Early Intensive Treatment

Initiating effective treatment early in the disease course in order to reduce relapse rate and the underlying inflammatory process may delay irreversible neurological damage and conversion to a secondary progressive course. The median time to conversion to a secondary progressive course is around fifteen years but can be shorter, especially in patients with aggressive disease [54]. The main goal of treatment must be to prevent accumulation of irreversible neurological disability and, in particular, to prevent conversion to a secondary progressive course [54].

However, clinical trials have short follow-up times, which might prevent detection of progression of disability and moreover disability worsening in these scenarios may reflect mainly disability accrual from relapses rather than true progression. Furthermore, extension phases have many biases that preclude long-term outcomes analysis, and moreover, clinical trials do not usually compare the escalation and EIT approach. Therefore, long-term outcomes that assess disability at five to ten years after treatment onset, and conversion to secondary progressive MS must be analyzed on real world experience.

Recent evidence from several observational studies, suggest that EIT provides a greater benefit than escalation treatment in decreasing the risk of developing SPMS and disability accrual at least in the medium-long term of 5 to 10 years [43,58,59,60,61,62].

A Danish observational study with 4 years follow-up showed that initial high efficacy treatment (with natalizumab, fingolimod, alemtuzumab, ocrelizumab or cladribine) compared to medium efficacy treatment in naive patients (using propensity score matched samples) was associated with a lower probability of 6-month confirmed EDSS worsening (16.7% vs. 30.1%, HR 0.53, *p* = 0.006) and of a first relapse (HR 0.50) up to 4 years. Although fingolimod was initially considered as HET, when reclassifying it as a medium DMT, comparable results as in the main study were found. When subgroup analysis of patients with high baseline disease activity was done, comparable results were found too [58].

Another real-life setting study showed long-term outcomes were more favorable following initial EIT (with natalizumab or alemtuzumab) vs. moderate-efficacy treatment (with interferons, glatiramer, teriflunomide, dimethyl fumarate and fingolimod). This cohort UK study that included 592 RRMS patients showed that EIT patients had a lower increase in EDSS score at 5 years than patients with the escalation approach (0.3 vs. 1.2, *p* = 0.002). Median time to sustained accumulation of disability was longer for the EIT, but no differences were found between the medium-efficacy DMT who escalated to high-efficacy DMT and the EIT group. However, 60% of those who escalated to HET had already developed disability accumulation while still receiving initial moderate-efficacy treatment before escalation. Despite this, patients that received initial EIT had a more active disease (pretreatment ARR 1.7 vs. 0.7), it was this group that had better long-term outcomes. Interestingly, age at onset of first DMT was also related to EDSS change at 5 years [59].

An observational study with data from the Swedish MS and MSBase registries, assessed the efficacy of HET (natalizumab, rituximab, ocrelizumab, alemtuzumab or mitoxantrone) started early (0–2 years from onset) compared to later (4–6 years from onset) using propensity score. Although this study did not compare efficacy with the escalation approach (and escalation was allowed in both groups), it proved that early HET within 2 years of disease onset is associated with lower hazard of disability progression and lower disability accumulation at 6 to 10 years of follow-up compared to late HET (mean EDSS score at 10 years: 2.3 vs. 3.5, *p* < 0.0001) [60].

An Italian multicentric study that analyzed long-term trajectories up to 10 years of EIT vs. escalation in naive RRMS, starting treatment within the first year of disease onset, demonstrated EIT strategy is more effective than escalation in controlling disability progression over time. In this study EIT included patients that received as first DMT fingolimod, natalizumab, mitoxantrone, alemtuzumab, ocrelizumab or cladribine while escalation group received the high efficacy DMT after at least 1 year of treatment with glatiramer acetate, interferons, azathioprine, teriflunomide or dimethyl fumarate. Patients were followed for 10 years, and propensity score matched for characteristics at first DMT before analysis, all having at least one relapse on the previous year and baseline mean EDSS of 2.6. EIT was significantly associated with lower disability progression measured by mean annual EDSS change compared to baseline value in all time points, including at 5 and 10 years. This effect not only persisted but continued to increase over time despite all patients in the escalation group being escalated to a higher-efficacy DMT [61].

Regarding conversion to SPMS, EIT has been associated with a lower risk of conversion than escalation. A multicentric cohort study with 1555 patients, using propensity score matching, showed that EIT (initial treatment with alemtuzumab, natalizumab and fingolimod) was associated with a lower risk of conversion to SPMS than initial treatment with interferons and glatiramer (HR 0.66, *p* = 0.046; with a 5-year absolute risk 7% vs. 12%, median follow-up, 5.8 years). However, the probability of conversion to SPMS was lower when interferons or glatiramer were started within 5 years of disease onset versus later, and when platform treatments were escalated to fingolimod, alemtuzumab or natalizumab within 5 years versus later (HR 0.76, *p* < 0.001, 5-year absolute risk 8% vs. 14%, median follow-up 5.3 years), which may reflect that when using the escalation approach, treatment failure must be promptly detected [43].

In relation to long-term outcomes of specifically induction treatment in observational studies, most of available data is mainly related to the older induction treatments such as mitoxantrone or cyclophosphamide compared to injectable medium-efficacy treatment. Prosperini et al., compared effects and safety of initial induction treatment with mitoxantrone or cyclophosphamide vs. escalation treatment starting with interferons in active RRMS (with median ARR of 2 and 60% of patients with baseline contrast enhancing lesions) using propensity score match, and found that a significantly lower proportion of patients of the induction group reached the milestone of EDSS 6 at 10 years (28% vs. 38.7%, HR 0.48, *p* = 0.024). Younger age was related with better outcomes in the induction group, and adverse events were more frequent after induction. Notably, although induction was not compared with initial sustained HET, some of the induction patients required escalation to fingolimod or other monoclonal antibodies, however, in a lower proportion than the escalation group (34.7% vs. 53.4%) [62].

Data related to newer induction treatments such as alemtuzumab or cladribine are usually analyzed together with other HET versus the escalation approach, but no observational studies of long-term outcomes comparing initial induction versus initial sustained HET are available [62].

Therefore, real-world data show that the escalation approach may be inadequate to prevent long-term outcomes compared to EIT and that initial EIT is related to a lower risk of developing SPMS and to lower disability accumulation at 5 and 10 years. However, evidence comparing long-term outcomes of induction treatment vs. sustained HET is scarce [58,59,60,62].

## 4. Future Evidence

To assess the effectiveness of EIT vs. Escalation, two pivotal clinical trials are currently ongoing; the TREAT-MS (TRaditional versus Early Aggressive Therapy for MS) trial and the DELIVER-MS (Determining the Effectiveness of earLy Intensive Versus Escalation approaches for the treatment of Relapsing-remitting MS) trial. The primary endpoint in TREAT-MS is time to sustained disability progression and the primary endpoint in DELIVER-MS is normalized whole brain volume loss from baseline to month 36. Interestingly both clinical trials consider the sphingosine-1-P modulators as medium efficacy treatments, while the first one considers cladribine as an EIT and the last one as a medium efficacy treatment [49,61,62].

This work has been supported by a grant from the Health Institute Carlos III: PI20/01446.

Authors should discuss the results and how they can be interpreted from the perspective of previous studies and of the working hypotheses. The findings and their implications should be discussed in the broadest context possible. Future research directions may also be highlighted.

## Figures and Tables

**Table 1 jpm-12-00119-t001:** Early Intensive Therapy. * There is not consensus regarding Cladribine and fingolimod (as some authors consider them HET and others not).

Early Intensive Therapy (EIT):	
Induction Treatment	Mitoxantrone Cyclophosphamide Stem Cell transplantation Alemtuzumab Cladribine *
Sustained High-Efficacy Treatment	Natalizumab Fingolimod * Anti-CD20 treatment

**Table 2 jpm-12-00119-t002:** Long-term outcomes of Early Intensive Treatments: Observational Studies.

Beneficial Long-Term Outcomes of EIT vs. Escalation		
Observational Studies	Follow-Up	Outcomes
Buron et al. [58]	4 years	Lower risk of 6 month EDSS worsening and of first relapse
Harding et al. [59]	5 years	Lower increase in EDSS Longer Median time to sustained accumulation of disability
He et al. [60]	6–10 years	Early HET within 2 years of disease onset is associated with lower hazard of disability progression and lower disability accumulation at 6 to 10 years of follow-up compared to late HET
Iaffaldano et al. [61]	10 years	Lower disability progression measured by mean annual EDSS change compared to baseline value in all time points, including at 5 and 10 years.
Brown et al. [46]	5.8 years	Lower risk of conversion to SPMS
Prosperini et al. [62]	10 years	Lower proportion of patients reached the milestone of EDSS 6 at 10 years

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
