# Peer review of "Escalation vs. Early Intense Therapy in Multiple Sclerosis"

_jpm, 2022, doi:10.3390/jpm12010119_

Round 1

Reviewer 1 Report

In this review entitled 'Escalation vs. early intense therapy in multiple sclerosis' authors have critically analyzed the two strategies (escalation or induction strategy) for treatment of multiple sclerosis. This is an interesting and well written review that deserves to be published in the journal 'Journal of Personalized Medicine'. But, before accepting it for publication authors should check the whole manuscript for linguistic, grammatical and typographical errors. They should also try to include tables and figures for data presentation to make the paper more appealing.

Author Response

Thank you for your evaluation. Now a new paragraph have been added to the introduction, and two new tables have been also uploaded.

"Continuing with the uncertainty surrounding MS, one of the main problems we face in treating MS is whether there is activation of the innate immune system, which has been linked to the progression of disability, from the onset of the disease or whether it occurs at some point as a result of dysregulation of the acquired immune system joint to the phenomenon of antigenic spread. There is currently no answer to this question, but very early initiation with induction drugs that reset the immune system, such as Alemtuzumab, have given the best results in terms of long-term progressive secondary progressive progression, which may indicate that intense early treatment could prevents activation of the innate immune system and triggering of the innate immune-dependent mechanisms responsible for the progression of the disability

The grammar have been reviewed in deep.

Many thanks for your suggestions

Reviewer 2 Report

The authors have done a great job in compiling all of the innumerable treatment schemes available for MS patients along with data from the various clinical trials. However, the authors could try to elaborate the introduction section by adding more physiological impact of MS. 

I found many typos which the authors should check and make corrections. The following line numbers have typos: 87,91,149,154,189,191,215,217,218,308.

Author Response

Thank you very much for your suggestions. Now the paper have been reviewed in deep.